# DrugTrail: Interpretable Drug Discovery via Structured Reasoning and Druggability-Tailored Preference Optimization

**Yurou Liu**[1,2,4,5]**, Mingyang Li**[2]**, Xinyuan Zhu**[2]**, Rui Jiao**[3]**, Yiming Dong**[2]**, Xinyu Tang**[1,4,5]**,
Yang Liu**[3]**, Jieping Ye**[2]**, Bing Su**[1,4,5] *]**, Zheng Wang**[2] *]

[1] Gaoling School of Artificial Intelligence, Renmin University of China, Beijing, China
[2] Tongyi Lab, Beijing, China
[3] Department of Computer Science and Technology, Tsinghua University, Beijing, China
[4] Beijing Key Laboratory of Research on Large Models and Intelligent Governance
[5] Engineering Research Center of Next-Generation Intelligent Search and Recommendation, MOE
yurouliu99@gmail.com

## Abstract

Machine learning promises to revolutionize drug discovery, but its "black-box" nature and narrow focus limit adoption by experts. While Large Language Models (LLMs) offer a path forward with their broad knowledge and interactivity, existing methods remain data-intensive and lack transparent reasoning. To address these issues, we present DrugTrail, an LLM-based framework for interpretable drug discovery that integrates structured reasoning trajectories with a Druggability-Tailored Preference Optimization (DTPO) strategy. It not only introduces structured reasoning traces to articulate the "how" and "why" behind its conclusions but also serve to guide task-specific reasoning pathways within the LLM's vast knowledge space, thereby enhancing its interpretability and reliability of its final outputs. Furthermore, based on the fact that optimizing for binding affinity alone does not equate to optimizing for druggability, DTPO explicitly moves beyond single-metric optimization and opens up a broader search space that balances affinity with other essential factors. Extensive experiments demonstrate the effectiveness of our approach and its generalizability to a wider range of biomolecular optimization domains, bridging the gap between LLM reasoning capabilities and trustworthy AI-assisted drug discovery.

## 1 Introduction

The rapid development of machine learning (ML) techniques has opened up tremendous potential to accelerate drug discovery (Schneider et al., 2019; Irwin & Shoichet, 2016). By automating virtual screening, molecular docking, and molecular editing, ML-based methods can greatly reduce both time and cost in early-stage drug development (Gorgulla et al., 2020; Lyu et al., 2019; Gentile et al., 2020). However, most existing AI tools remain constrained by narrow, domain-specific knowledge and tend to operate as black boxes, offering limited transparency in their decision-making processes (Yu et al., 2024). Such limited interpretability impedes collaborations with domain experts, constraining effective analysis and refinement within real biomedical workflows (Li et al., 2023).

Recently, large language models (LLMs) have emerged as general-purpose systems with broad cross-domain knowledge and the capability to generate human-readable explanations (Edwards et al., 2022; 2021). While these features position them as promising candidates for assisting drug discovery, most existing LLM-based methods require large-scale biologically relevant data for pre-training (Xia et al., 2025). This requirement not only poses challenges due to the scarcity of high-quality biological datasets, but also entails substantial computational costs. Moreover, current methods typically focus on producing final predictions without explicitly revealing the intermediate reasoning steps. In a high-stakes field like drug discovery, understanding the reasoning process is as important as the

---

*Correspondence to: Bing Su <bingsu@ruc.edu.cn>, Zheng Wang <wz388779@alibaba-inc.com>.

results themselves, as it improves interpretability and allows domain experts to contextualize and adapt AI-generated hypotheses. Notably, prior work has demonstrated that instead of building domain-specific models through costly pre-training, reinforcement learning (RL) can unlock domain-relevant knowledge already embedded in general-purpose LLMs, as evidenced in tasks such as mathematics and programming (Yu et al., 2025). Plus, a modest amount of supervised fine-tuning (SFT) is sufficient to bootstrap their reasoning capabilities (Guo et al., 2025a). Therefore, exploring RL approaches with reasoning holds significant potential for advancing LLM-based drug discovery.

However, enabling a reasoning-enhanced RL framework for drug discovery faces two core challenges. First, the reasoning process should follow biochemistry-informed modes of thinking. Current datasets, however, lack structural reasoning trajectories that can serve as guidance for learning domain-specific reasoning patterns. Second, designing an appropriate reward remains a critical bottleneck. Previous methods often rely on affinity scores for guidance, but directly optimizing binding affinity does not necessarily translate into drug-likeness in real-world scenarios. Factors such as binding duration, synthetic accessibility, and potential off-target inhibition also play essential roles in determining druggability (Zhang et al., 2025). Therefore, an effective RL framework must incorporate a more comprehensive reward design that better aligns with practical pharmacological criteria.

In this work, we introduce DRUGTRAIL, an LLM-based framework for interpretable drug discovery. To address the scarcity of explicit reasoning data, we leverage existing pocket–ligand pairs and prompt the language model to complete multi-dimensional reasoning trajectories under biochemistry-informed rules. Based on the resulting pocket–reasoning–ligand data, we apply supervised fine-tuning (SFT) to guide the model toward generating structural and interpretable reasoning. Subsequently, rather than relying on time-consuming affinity scores as optimization targets, DRUGTRAIL incorporates a Druggability-Tailored Preference Optimization (DTPO) strategy. This strategy jointly considers the similarity between generated candidates and known drug-like molecules, as well as rule-based indicators of intrinsic druggability, thereby enabling efficient online computation while maintaining a strong connection between rewards and drug-likeness. To sum up, our contributions are as follows:

• **Interpretable Framework:** We design DRUGTRAIL, a novel paradigm for drug design, in which lightweight reinforcement learning activates LLMs' capability to generate interpretable, high-quality biomolecular candidates.

• **Druggable Optimization:** We propose Druggability-Tailored Preference Optimization (DTPO), a computationally efficient, online-computable preference scheme with a hybrid reward coupling candidate-drug similarity with rule-based intrinsic drug-likeness indicators. We also release a large-scale bioactive reference dataset to facilitate future research.

• **Generalizable Performance:** Extensive experiments across de novo pocket-based drug design, small-molecule editing, and protein optimization demonstrate consistently strong performance and transferability, paving the way toward more interpretable and widely adoptable AI tools in pharmaceutical research.

## 2 METHOD

In this section, we introduce DRUGTRAIL, an interpretable framework for drug discovery designed to overcome two primary limitations of existing methods: (1) the absence of transparent reasoning processes that align with established principles of medicinal chemistry, and (2) the constrained optimization focus on binding affinity , which often neglects other critical factors for druggability (Pantsar & Poso, 2018).[1] As illustrated in Fig. 1, DRUGTRAIL is composed of two core modules. First, a Clinical Chemistry-Informed Reasoning (CCIR) module generates structured and interpretable reasoning trajectories. Second, a Druggability-Tailored Preference Optimization (DTPO) strategy optimizes for a comprehensive set of drug-like properties, moving beyond single-metric objectives.

---

[1] Generally, binding affinity refers to the strength of the interaction between a ligand and its biological target, while druggability describes whether a molecule has the properties and potential required to be developed into a safe and effective therapeutic drug.

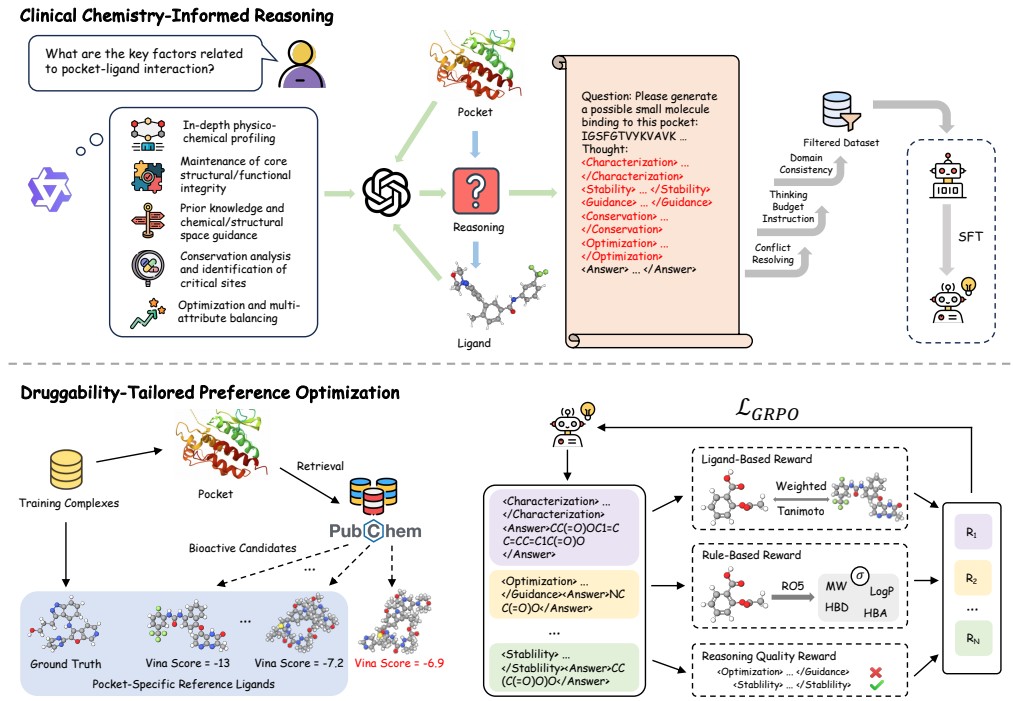

Figure 1: The overall framework of DRUGTRAIL. The upper part illustrates the Clinical Chemistry–Informed Reasoning (CCIR) process. The lower part presents the workflow of Druggability-Tailored Preference Optimization (DTPO).

## 2.1 CLINICAL CHEMISTRY-INFORMED REASONING (CCIR)

### 2.1.1 STRUCTURED REASONING DIMENSION CONSTRUCTION

To address the limited interpretability of current language models in drug discovery, our primary objective is to construct a reasoning framework that emulates the decision-making processes of medicinal chemists. The goal is to enhance the credibility and reliability of the model's predictions by rendering its reasoning process transparent and concordant with established scientific principles. A central challenge in this endeavor is the definition of structured reasoning dimensions capable of effectively guiding the model's inference pathway.

Drawing upon the demonstrated success of Large Language Models (LLMs) in complex reasoning tasks such as mathematics and code generation, we hypothesize that general-purpose LLMs possess substantial latent knowledge relevant to specialized domains. To elicit this knowledge, we designed a structured prompt to query a foundational model, Qwen3-235B-A22B, thereby extracting common reasoning perspectives pertinent to clinical drug discovery. This process yielded five core reasoning dimensions: (1) ***In-depth physicochemical profiling***; (2) ***Maintenance of core structural and functional integrity***; (3) ***Prior knowledge and chemical / structural space guidance***; (4) ***Conservation analysis and identification of critical sites***; (5) ***Optimization and multi-attribute balancing***. Together, these five dimensions form the foundation of our reasoning framework, which we further operationalize in subsequent sections.

### 2.1.2 TRAJECTORY GENERATION FROM MULTI-OBJECTIVE QUALITY CONTROL

To enable the model to explicitly learn these clinical chemistry-informed thinking modes, we constructed a high-quality dataset for supervised fine-tuning. We sampled data from the training split of the Cross-Docked2020 dataset (Francoeur et al., 2020), where each sample comprises a protein binding pocket and its corresponding interacting ligand. For each sample, we formulated a question and a corresponding answer that contains a reasoning trajectory structured according to the five aforementioned dimensions. Specifically, we instructed the large language model to complete the reasoning process that connects the pocket to its

interacting ligand. To ensure clarity and consistency, the reasoning output was formatted with special tokens assigned to each dimension: `<Characterization>...</Characterization>`, `<Stability>...</Stability>`, `<Guidance>...</Guidance>`, `<Conservation>...</Conservation>`, and `<Optimization>...</Optimization>`. Through this design, we obtained the initial version of a comprehensive Pocket–Reasoning–Ligand structural dataset. To further ensure the quality and reliability of these generated trajectories, we implemented a three-stage data filtering pipeline.

**Conflict Resolving.** To enhance the logical coherence of the reasoning trajectories, we implemented a conflict resolution protocol. For each input sample, multiple reasoning candidates were generated using stochastic decoding or an ensemble of models. These diverse trajectories were then evaluated by a powerful adjudicator LLM to assess their internal consistency. Trajectories containing logical fallacies, self-contradictions, or conflicting biochemical claims were identified and discarded, ensuring that only the most robust and logically sound reasoning paths were retained for fine-tuning.

**Thinking Budget Instruction.** We hypothesize that effective reasoning should be both comprehensive and concise, avoiding unnecessary verbosity or overly simplistic explanations. To enforce this, we introduce a "thinking budget" constraint. This is operationalized through a filtering mechanism that penalizes reasoning trajectories whose lengths fall outside a predefined range, thereby discouraging both excessively long and perfunctory outputs. Trajectories that are identified as outliers in length are discarded, guiding the model to learn a more focused and efficient reasoning style. Further details on the implementation of this heuristic are provided in the Appendix B.

**Domain Consistency.** To ensure that the model-generated reasoning aligns with the cognitive patterns of human experts, we employ a domain consistency check grounded in an adversarial validation framework. We curate a small set of "golden" reasoning trajectories authored by domain experts. These expert examples are then combined with the LLM-generated trajectories, and a separate LLM serves as a classifier to reject the generated trajectories that deviate significantly from the expert-authored ones. By selectively retaining these consistent samples, we ensure the final training data is closely aligned with established expert knowledge and thought processes.

### 2.1.3 DOMAIN KNOWLEDGE ENHANCEMENT FROM SUPERVISED FINE-TUNING

Following the multi-stage data curation and quality control pipeline described above, we obtained a corpus of 2,000 high-quality annotated samples for domain-specific Supervised Fine-Tuning (SFT). Each sample consists of a problem statement (question) and a structured expert-informed solution (answer). The answer is organized as a structured reasoning trajectory consisting of six parts: five clinical chemistry-informed analytical aspects introduced in § 2.1.2 and the final answer segment `<Answer>, </Answer>` containing the final analysis and the proposed molecule in canonical SMILES notation (Weininger, 1988) (*e.g.* `<Answer>Based on the above analysis, the final answer is \boxed{CCO}</Answer>`). Given the collected SFT dataset, we adopt a standard cross-entropy loss to align the base model with the structural reasoning format and medicinal chemistry knowledge. This SFT phase provides the model with a structured cognitive prior, enhancing both its interpretability and reliability in downstream drug design tasks.

## 2.2 DRUGGABILITY-TAILORED PREFERENCE OPTIMIZATION (DTPO)

### 2.2.1 GROUP RELATIVE POLICY OPTIMIZATION (GRPO)

GRPO is a reinforcement learning framework that eliminates value models by estimating *group-wise* relative advantages. Specifically, for each question $q \sim \mathcal{D}$, we first sample a group of $G$ responses $\{o_i\}_{i=1}^{G}$ from the old policy model $\pi_{\text{old}}$. The policy model $\pi_\theta$ is then optimized using a clipped surrogate objective with a KL regularization term to enhance training stability, given by,

$$\mathcal{L}_{\text{GRPO}}(\theta) = -\mathbb{E}_{q\sim\mathcal{D},\, \{o_i\}_{i=1}^{G}\sim\pi_{\text{old}}(\cdot|q)}$$

$$\left[ \frac{1}{G}\sum_{i=1}^{G}\frac{1}{|o_i|}\sum_{t=1}^{|o_i|}\left( \min(r_{i,t}\,A_{i,t},\ \text{clip}(r_{i,t}, 1-\varepsilon,\ 1+\varepsilon)\,A_{i,t}) - \beta D_{\text{KL}}(\pi_\theta||\pi_{\text{ref}}) \right) \right], \quad (1)$$

where $r_{i,t} = \pi_\theta(o_{i,t} \mid q, o_{i,<t}) / \pi_{\theta_{\text{old}}}(o_{i,t} \mid q, o_{i,<t})$ is the importance weights for the $t$-th token $o_{i,t}$. The advantage is estimated from group-normalized rewards $\{R_i\}_{i=1}^G$,

$$A_{i,t} = \frac{R_i - \text{mean}\big(\{R_i\}_{i=1}^G\big)}{\text{std}\big(\{R_i\}_{i=1}^G\big)}. \tag{2}$$

### 2.2.2 Construction of Preference Data

To steer the generative model towards regions of the chemical space with superior druggability, we implemented a hybrid ligand- and structure-based data curation pipeline. The objective was to construct a preference dataset that reflects a nuanced combination of established bioactivity and structural compatibility, transcending simple binding affinity metrics.

**Target-Centric Retrieval of Bioactive Compounds.** For each protein pocket in the CrossDocked dataset, we first identified its biological target, and then systematically queried the PubChem database (Kim et al., 2016) to compile a comprehensive set of molecules with experimentally validated activity against this target. This step enriches our initial candidate pool with biologically relevant chemical scaffolds.

**Structure-Based Virtual Screening and Validation.** The retrieved bioactive compounds subsequently underwent virtual screening via molecular docking with AutoDock Vina (Trott & Olson, 2010). For each pocket, we retained the ground truth ligand and candidate molecules from the top 20 docking results with Vina score lower than -7. This strategy ensures that every molecule in our final reference set exhibits strong binding potential while also expanding the structural diversity beyond the single co-crystallized ligand in the original dataset.

This curated set provides a rich collection of positive examples that forms the empirical foundation for our reward structure.

### 2.2.3 Reward Design

To effectively guide the model's optimization, we designed a multi-component reward function that incorporates principles of medicinal chemistry and structured reasoning. The total reward is a composite of three components, which are detailed as follows:

**Ligand-based Reward with Adaptive Ranking.** Based on the expanded reference dataset, this component is defined as:

$$R_{\text{ligand}}(m) = \sum_{i=1}^N \gamma^{\text{rank}_i} \text{Tanimoto}(m, r_i) \tag{3}$$

Structural similarity between a *de novo* molecule $m$ and each reference compound $r_i$ is quantified via the Tanimoto coefficient (Chen & Reynolds, 2002). References, pre-ranked by docking affinity, are weighted by $\gamma^{\text{rank}_i}$ using an attenuation factor $\gamma = 0.95$, thus exponentially prioritizing resemblance to top-ranked ligands and rewarding molecules inheriting their key structural features.

**Rule-Based Reward via Soft Lipinski's Rule of Five.** To encourage the generation of molecules with favorable physicochemical properties, we implemented Lipinski's Rule of Five (RO5) (Lipinski, 2004a) as a soft scoring function. Each of the four RO5 properties—molecular weight (MW), octanol-water partition coefficient (LogP), hydrogen bond donors (HBD), and hydrogen bond acceptors (HBA)—is mapped to an individual score in $(0, 1)$. The soft score $s(x)$ for a property value $x$ with threshold $t$ can be computed using a bounded linear or sigmoid decay, such as:

$$s(x) = \frac{1}{1 + \exp(k(x - t))} \tag{4}$$

where $k$ is a steepness parameter. The overall RO5 soft score for molecule $m$ is the average of the four property scores:

$$R_{\text{rule}}(m) = \frac{s_{\text{MW}} + s_{\text{LogP}} + s_{\text{HBD}} + s_{\text{HBA}}}{4} \tag{5}$$

where $R_{\text{rule}}(m) \in (0, 1)$, with higher values indicating better drug-likeness.

Table 1: Results of interaction analysis.

| Model | Vina Dock | | | | PLIP Interaction | | | |
|---|---|---|---|---|---|---|---|---|
| | E ↓ | IMP (%) ↑ | MPBG (%) ↑ | LBE ↑ | $JSD_{OA}$ ↓ | $MAE_{OA}$ ↓ | $JSD_{PP}$ ↓ | $MAE_{PP}$ ↓ |
| **Baselines (w Reasoning)** | | | | | | | | |
| DeepSeek-R1 | -0.36 | 0.2 | -101.33 | 0.0814 | 0.3521 | 0.5076 | 0.7041 | 0.8974 |
| Qwen3-235B-A22B | -1.21 | 0.82 | -91.26 | 0.0576 | 0.3273 | 0.5023 | 0.6815 | 0.9428 |
| QwQ-32B | 10.23 | 0.00 | -282.14 | -0.4447 | 0.6674 | 0.9723 | 0.7912 | 1.6940 |
| Qwen3-4B-Thinking | 11.81 | 0.00 | - | - | 0.7149 | 1.0461 | 0.8066 | 1.5482 |
| **Baselines (w/o Reasoning)** | | | | | | | | |
| DeepSeek-V3 | 5.91 | 0.09 | -176.43 | -0.3421 | 0.4136 | 0.5276 | 0.7415 | 1.3874 |
| Qwen3-235B-A22B-non-thinking | 7.36 | 0.00 | -241.23 | -0.3681 | 0.5021 | 0.8710 | 0.7123 | 1.2136 |
| Llama-3.1-8B | 21.44 | 0.00 | - | - | 0.7792 | 1.1613 | 0.8159 | 1.7192 |
| **Qwen3-1.7B** | | | | | | | | |
| Base | 9.77 | 0.00 | - | - | 0.6762 | 0.9572 | 0.7535 | 1.5012 |
| + CCIR | -3.06 | 8.59 | -62.33 | 0.2599 | 0.0677 | 0.1454 | 0.2311 | 0.5609 |
| **+ CCIR + DTPO** | -6.65 | 33.48 | 0.83 | 0.2892 | 0.0428 | 0.1197 | 0.2135 | 0.4927 |
| **Qwen3-4B** | | | | | | | | |
| Base | 12.49 | 0.00 | - | - | 0.7158 | 0.9791 | 0.7620 | 1.4099 |
| + CCIR | -3.09 | 11.18 | -61.58 | 0.2621 | 0.0603 | 0.1372 | 0.2270 | 0.5341 |
| **+ CCIR + DTPO** | -6.79 | 38.19 | 2.59 | 0.3361 | 0.0465 | 0.1263 | 0.2044 | 0.4753 |
| **Qwen3-8B** | | | | | | | | |
| Base | 11.80 | 0.01 | -268.57 | -0.5578 | 0.6508 | 0.8891 | 0.7328 | 1.4753 |
| + CCIR | -3.10 | 10.93 | -60.91 | 0.2653 | 0.0625 | 0.1410 | 0.1868 | 0.5269 |
| **+ CCIR + DTPO** | **-6.82** | **41.01** | **6.47** | **0.3407** | **0.0395** | **0.0922** | **0.1782** | **0.4506** |

**Reasoning Quality Reward.** To enforce the correct use of the structured reasoning format, we designed a token-completeness reward. For each of the six predefined pairs of special tokens (e.g., `<Characterization>`...`</Characterization>`) that is correctly and completely included in the generated output, the model receives an additive reward. The total reasoning quality reward for a given trace is:

$$R_{\text{reasoning}}(\text{trace}) = 0.1 \times N_{\text{pairs}}(\text{trace}) \tag{6}$$

where $N_{\text{pairs}}(\text{trace}) \in \{0, 1, \cdots, 6\}$ is the count of correctly matched token pairs. This incentivizes the inclusion of all reasoning dimensions, ensuring both completeness and structural clarity.

### 2.2.4 TRAINING OBJECTIVE

The final training objective of DRUGTRAIL is to optimize the policy model $\pi_\theta$ using the GRPO framework. The core of this optimization lies in the design of the reward signal, which guides the model's learning process. The total reward, $R_{total}$, used to compute the advantages within the GRPO loss function is a weighted linear combination of the three reward components detailed in § 2.2.3:

$$R_{\text{total}} = w_{\text{ligand}} R_{\text{ligand}} + w_{\text{rule}} R_{\text{rule}} + w_{\text{reasoning}} R_{\text{reasoning}} \tag{7}$$

where the coefficients $w_{\text{ligand}}$, $w_{\text{rule}}$, and $w_{\text{reasoning}}$ are non-negative hyperparameters that balance the contribution of each component to the total reward. These weights are crucial for tuning the model's behavior, allowing us to modulate the emphasis between generating structurally relevant molecules, ensuring their druggability, and maintaining a coherent, structured reasoning process.

## 3 EXPERIMENTS

### 3.1 SETTINGS

We employ the CBGBench framework (Lin et al., 2024) to evaluate our DRUGTRAIL on Cross-docked2020 dataset (Francoeur et al., 2020) across three dimensions: substructure, chemical properties and interaction which are detailed in Appendix. C.1.1. We use its established splits with 100 complexes for testing (Peng et al., 2022). Noted that our evaluation excludes the geometric dimension as our method does not perform direct 3D generation. The baselines are detailed in Appendix. C.1.2.

Table 2: Results of chemical property analysis. It should be noted that for drug candidates, optimal LogP values are typically in the range of -0.4 to 5.6, showing no preference toward either higher or lower values within this interval (Lin et al., 2024).

| Method | QED ↑ | LogP | SA ↑ | LPSK ↑ |
|---|---|---|---|---|
| **Qwen3-1.7B** | | | | |
| Base | 0.12 | 4.17 | 0.23 | 4.08 |
| + CCIR | 0.31 | 2.56 | 0.40 | 4.37 |
| **+ CCIR + DTPO** | 0.55 | 1.72 | 0.70 | **4.82** |
| **Qwen3-4B** | | | | |
| Base | 0.16 | 4.10 | 0.24 | 4.09 |
| + CCIR | 0.30 | 2.58 | 0.40 | 4.38 |
| **+ CCIR + DTPO** | **0.57** | 1.71 | 0.69 | **4.82** |
| **Qwen3-8B** | | | | |
| Base | 0.17 | 4.12 | 0.28 | 4.02 |
| + CCIR | 0.31 | 2.58 | 0.43 | 4.38 |
| **+ CCIR + DTPO** | **0.57** | 1.71 | **0.72** | 4.81 |

Table 3: Results of substructure analysis. Functional groups are evaluated based on 25 categories identified using the EFG (Salmina et al., 2015). JSD and MAE measure distribution divergence and occurrence frequency differences between generated and reference substructures.

| Method | Atom Type | | Ring Type | | Function Group | |
|---|---|---|---|---|---|---|
| | JSD ↓ | MAE ↓ | JSD ↓ | MAE ↓ | JSD ↓ | MAE ↓ |
| **Qwen3-1.7B** | | | | | | |
| Base | 0.2786 | 1.7654 | 0.5528 | 0.5476 | 0.5872 | 0.0981 |
| + CCIR | 0.1423 | 1.0620 | 0.3015 | 0.2931 | 0.3457 | 0.0677 |
| **+ CCIR + DTPO** | 0.0689 | 0.3113 | 0.2014 | 0.1789 | 0.2476 | 0.0501 |
| **Qwen3-4B** | | | | | | |
| Base | 0.2676 | 1.5654 | 0.5121 | 0.5039 | 0.5284 | 0.0814 |
| + CCIR | 0.0897 | 0.3362 | 0.2311 | 0.2059 | 0.2718 | 0.0518 |
| **+ CCIR + DTPO** | 0.0453 | 0.3065 | 0.1681 | 0.1219 | 0.2043 | 0.0442 |
| **Qwen3-8B** | | | | | | |
| Base | 0.2612 | 1.5472 | 0.5040 | 0.4929 | 0.5015 | 0.0787 |
| + CCIR | 0.0698 | 0.3107 | 0.2188 | 0.1958 | 0.2399 | 0.0489 |
| **+ CCIR + DTPO** | **0.0401** | **0.2989** | **0.1471** | **0.1108** | **0.1869** | **0.0303** |

## 3.2 MAIN RESULTS

The interaction analysis results are presented in Tab. 1. Across all model sizes (Qwen3-1.7B, 4B, 8B), our method consistently achieves substantial improvements especially in docking energy (E) and the proportion of superior binders (IMP). Incorporating the reasoning-based CCIR module markedly activates the models' capacity to generate pocket-specific ligands, while the addition of DTPO delivers even more pronounced gains, e.g., reducing E to -6.65 $\sim$ –6.82 and raising IMP to 33–41%, far surpassing all baselines.

Beyond interaction-based metrics, we further evaluate the chemical and structural properties of the generated candidates in Tab. 2 and 3, respectively. Incorporating DTPO consistently improves QED (0.55–0.57) and SA (0.69–0.72), while maintaining high LPSK scores, indicating candidates with better synthetic feasibility and rule-based druggability. Moreover, DTPO leads to a closer match to the reference set in functional group distribution, suggesting that it not only optimizes general chemical features but also implicitly captures pocket-specific structural motifs essential for ligand binding, thereby enhancing the plausibility and pharmacological relevance of the generated molecules.

## 3.3 GENERALIZATION EXPLORATION

To assess the potential of our method to generalize across a broader range of biomolecular domains, we further conduct exploratory experiments in this section on two representative tasks: zero-shot small-molecule editing and large-molecule protein optimization, aiming to provide initial empirical evidence of its broader applicability beyond the specific settings in which it is originally developed.

### 3.3.1 MOLECULE EDITING

**Experimental setup.** We first examine the zero-shot performance of DRUGTRAIL on small-molecule editing. Following (Liu et al., 2024b), we use 200 molecules sampled from the ZINC dataset (Irwin et al., 2012) for evaluation. Note that for this task, we directly use the same model trained on CrossDocked2020 without additional post-training. See Appendix C.2 for more details.

**Results.** As shown in Tab. 5, across all configurations, the performance

Table 4: Results of protein editing analysis (GFP optimization). Noted that higher diversity and novelty are not equivalent to better performance, but provide insight into the exploration and exploitation trade-offs of different methods (Kirjner et al., 2023).

| Method | Medium Difficulty | | | Hard Difficulty | | |
|---|---|---|---|---|---|---|
| | Fitness ↑ | Diversity | Novelty | Fitness ↑ | Diversity | Novelty |
| **Qwen3-1.7B** | | | | | | |
| Base | 0.07 | 112 | 201 | 0.05 | 94 | 156 |
| DRUGTRAIL | 0.53 | 22.3 | 26.1 | 0.54 | 18.7 | 27.8 |
| **Qwen3-4B** | | | | | | |
| Base | 0.08 | 113 | 129 | 0.07 | 131 | 146 |
| DRUGTRAIL | 0.60 | 19.4 | 29.7 | **0.57** | 20.5 | 18.9 |
| **Qwen3-8B** | | | | | | |
| Base | 0.08 | 142 | 187 | 0.1 | 125 | 193 |
| DRUGTRAIL | **0.62** | 13.1 | 7.9 | **0.57** | 17.6 | 8.4 |

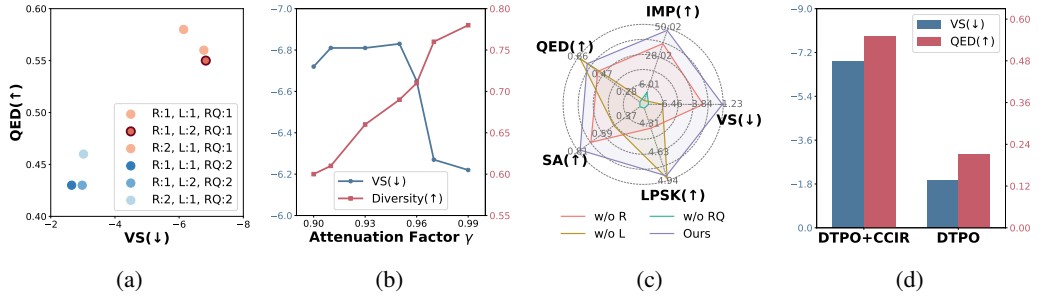

Figure 2: Visualization of ablation studies. **a.** Weights of different components of reward function. The setting adopted in DRUGTRAIL is highlighted in red. VS denotes the Vina Score (dock), R, L, RQ denote the Rule-Based, Ligand-Based, and Reasoning Quality Reward, respectively. The top-right area in the figure represents the ideal region. **b.** Value of attenuation factor $\gamma$ in the Ligand-Based Reward. **c.** Components of reward function. **d.** Necessity of the CCIR stage.

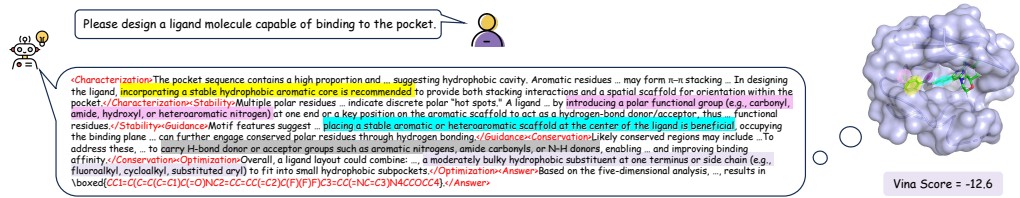

Figure 3: Visualization of a candidate molecule generated by DRUGTRAIL along with its corresponding reasoning process. Our reasoning process aligns well with the structure of the generated candidate molecule and enables successful, high-quality docking with the corresponding binding pocket. Text segments and molecular substructures highlighted in the same color represent matched pairs.

gap between DRUGTRAIL and the Base method remains significant, with gains exceeding 30 percentage points in most scenarios. It suggests that the interpretable design, bio-active chemical space construction, and rule-based constraints have guided the model to learn editing mechanisms that effectively capture transferable structure–function relationships, enabling robust zero-shot adaptation to molecule editing tasks.

### 3.3.2 PROTEIN EDITING

**Experimental setup.** In addition, we evaluate the generalizability of this interpretable training paradigm to macromolecule protein optimization tasks. Following Kirjner et al. (2023), we perform an evaluation on the Green Fluorescent Protein (GFP), with details provided in Appendix C.3.2.

**Results.** From Tab. 4, we can see that across all model scales, DRUG-TRAIL achieves markedly higher fitness scores than the Base method,

Table 5: Results of molecule editing analysis with performance assessed by the hit ratio of the property change ($\uparrow$).

| Method | Water Solubility | | Drug-likeness | | Permeability | | More Hydrogen Bond | |
|---|---|---|---|---|---|---|---|---|
| | More | Less | More | Less | More | Less | Acceptors | Donors |
| **Qwen3-1.7B** | | | | | | | | |
| Base | 36.33 | 40.33 | 32.17 | 35.17 | 29.50 | 21.67 | 10.67 | 13.83 |
| DRUGTRAIL | 75.67 | 69.83 | 50.17 | 66.83 | 51.67 | 61.83 | 39.50 | 41.00 |
| **Qwen3-4B** | | | | | | | | |
| Base | 40.50 | 42.33 | 33.67 | 35.17 | 30.33 | 22.67 | 11.67 | 14.17 |
| DRUGTRAIL | 76.33 | 70.17 | **53.50** | 71.00 | 52.33 | 64.17 | 41.83 | **43.50** |
| **Qwen3-8B** | | | | | | | | |
| Base | 38.33 | 42.83 | 34.50 | 33.33 | 31.50 | 23.17 | 10.50 | 15.67 |
| DRUGTRAIL | **76.50** | **72.50** | 52.00 | **71.17** | **54.67** | **66.83** | **44.33** | 41.83 |

with improvements from below 0.1 to above 0.5 in both difficulty levels. While DRUGTRAIL's diversity and novelty scores are lower than those of the Base method, the substantially higher fitness values reflect a stronger exploitation of beneficial sequence modifications, rather than broad but low-quality exploration. It suggests that the model leverages the structured thinking mode to focus the search within regions of the sequence space that are more likely to yield functional improvements, enabling more precise optimization orientation.

# 4 ANALYSIS

## 4.1 ABLATION STUDIES

**Weights of Different Components of Reward Function.** Determining appropriate weight assignments for each component of the reward function is critical to the reinforcement learning process. As shown in Fig. 2a, when a higher weight is assigned to the Rule-Based Reward, the QED score of the generated candidate molecules tends to increase, while their binding affinity decreases. Conversely, increasing the weight of the Ligand-Based Reward produces the opposite trend. It highlights the need to balance and trade off between the two performance aspects. Based on this analysis, we select a R:L:RQ=1:2:1 weighting scheme that lies within the optimal region.

**The Attenuation Factor $\gamma$.** In DTPO preference data construction, each protein pocket is paired with experimentally validated bioactive candidates ranked by docking scores. The model is guided to favor higher-ranked compounds through the attenuation factor $\gamma$. As shown in Fig. 2b, larger $\gamma$ increases molecular diversity. While diversity remains valuable, in this setting we prioritize binding affinity and thus select $\gamma = 0.95$, corresponding to its extremum.

**Components of Reward Function.** As illustrated in Fig. 2c, we evaluate five aspects reflecting biological efficacy and chemical properties to analyze the three reward components proposed in § 2.2.3. We first notice that removing the Reasoning Quality Reward leads to a marked drop across all aspects, indicating that the structured biological reasoning mode is crucial for RL. Considering the other two rewards, the Ligand-Based Reward plays a key role in enhancing interaction awareness, as its removal results in substantial declines in IMP and VS. Complementarily, the Rule-Based Reward primarily improves drug-likeness, with its removal causing a pronounced drop in LPSK. Together, these results reinforce the notion that effective drug discovery requires balancing binding affinity and drug-likeness, neither of which can be compromised.

**CCIR.** We further investigate the role of the Clinical Chemistry-Informed Reasoning (CCIR) process in enhancing the model's capability for drug discovery tasks. As illustrated in Fig. 2d, removing the CCIR process and directly applying DTPO resulted in a significant drop in performance with respect to both binding affinity and drug-likeness. Once again this confirms the importance of incorporating an interpretable drug discovery prior into our method.

**Geometry-related Reasoning Dimensions.** Geometric information is essential for drug design. We apply five dimensions in constructing the reasoning trajectory, among which three implicitly embed geometric considerations. We refer to those as the Geometry-related Reasoning Dimensions (GRD): **1.** *Maintenance of core structural and functional integrity* captures the scaffold and the orientation of groups that form hydrogen bonds, salt bridges, and $\pi$–$\pi$ stacking; **2.** *Prior knowledge and chemical/structural space guidance* utilizes known binding patterns within the target family to guide candidate structures; **3.** *Conservation analysis and identification of critical sites* leverages homology and pocket conservation to locate key residues and constrain binding modes. As shown in Tab. 6, removing these dimensions significantly reduces the binding quality between the generated ligands and the receptor pocket. This indicates that the geometric analysis introduced by GRD provides a substantial positive information gain for drug design. GRD acts as soft constraints on pocket shape and interaction geometry, steering the model toward candidates that better satisfy the pocket's spatial constraints and known interaction motifs. Consequently, DRUGTRAIL can maintain a certain degree of geometric consistency and druggability without explicit 3D coordinate input.

**Affinity Evaluation.** Accurate and rigorous affinity evaluation is essential for improving and validating model performance. We use multiple affinity prediction methods to screen and rank the preference dataset composed of bioactive compounds, and we evaluate with multiple metrics. Specifically, we build three variants: DRUGTRAIL–Vina (ranked by Vina score), DRUGTRAIL–Boltz2 (ranked by Boltz2 (Passaro et al., 2025) predicted affinity), and DRUGTRAIL–SIU (ranked by pIC50 labels provided in the SIU (Huang et al., 2024) dataset from wet-lab assays). As shown in Tab. 7, the model is robust across all combinations of "screening strategy × evaluation metric". It is worth noting that in the DRUGTRAIL–SIU setting we obtain 78 Uniprot IDs by taking the intersection of our training set and the SIU dataset, in which the preference dataset is only about half the size of the other settings (some compounds are not included in SIU), yet the results remain competitive. This indicates that DRUGTRAIL does not depend on certain affinity evaluation tools but serves as a generalizable framework.

Table 6: Ablation study on GRD. Boltz2 Affinity refers to pIC50 (kcal/mol). The two metrics do not exhibit a strong linear relationship.

| Method | Vina Score ↓ | Boltz2 Affinity ↑ |
|---|---|---|
| DRUGTRAIL (w/o GRD) | -4.61 | 3.99 |
| DRUGTRAIL | **-6.79** | **6.68** |

Table 7: Ablation study on different binding affinity evaluation methods.

| Method | Vina Score ↓ | Boltz2 Affinity ↑ |
|---|---|---|
| DRUGTRAIL-Boltz2 | -6.71 | **6.75** |
| DRUGTRAIL-SIU | -6.52 | 6.47 |
| DRUGTRAIL-Vina | **-6.79** | 6.68 |

## 4.2 CASE STUDIES

We randomly sample the 3TV6 pocket from the validation set to show our interpretable drug design process, which is illustrated in Fig. 3 and detailed in Appendix E. The reasoning analysis of the selected pocket reveals dominant hydrophobic and aromatic residues alongside discrete polar hot spots, suggesting requirements for an aromatic core to engage $\pi-\pi$ stacking and a polar functional group for directional hydrogen bonding. After docking the generated candidate into the pocket, we find that these analyses lead to features such as the phenyl ring stacking with aromatic side chains and the carbonyl group oriented toward a polar region. This alignment between reasoning and observed binding illustrates the mechanism underlying the interaction. With such an interpretable reasoning process, we can understand "why" and "how" the model designs and this binding occurs.

## 5 RELATED WORK

**LLMs for Drug Discovery.** Large language models (LLMs) have attracted growing attention in drug discovery across diverse domains and tasks. Existing approaches to adapting LLMs for drug discovery can be categorized into three main types: training models from scratch for specific domains (Chen et al., 2024; Hayes et al., 2025; Zhu et al., 2025; Nguyen et al., 2024; Liu et al., 2023; Luo et al., 2022; Taylor et al., 2022), fine-tuning general-purpose LLMs (Chaves et al., 2024; Lv et al., 2025; Wu et al., 2026; 2025), and incorporating pretrained scientific encoders using adapters (Liang et al., 2023; Luo et al., 2023; Li et al., 2025). More recently, unified frameworks like NatureLM (Xia et al., 2025) have been pretrained on large mixed-domain corpora to achieve broad coverage. Despite their promise, these models are still data-intensive and prioritize final predictions over interpretable reasoning, highlighting the need for more efficient and interpretable methods. We provide more discussions in Appendix D.

**Post-Training for LLM Reasoning.** The powerful reasoning capabilities of LLMs have been significantly enhanced through post-training techniques. Supervised Fine-Tuning (SFT) (Yang et al., 2024; Min et al., 2024) has played an important role by training models on high-quality, step-by-step reasoning sequences, improving their ability to generate coherent Chain-of-Thought (CoT) reasoning. More recently, Reinforcement Learning (RL) (Guo et al., 2025a; Yu et al., 2025; Yue et al., 2025; Wang et al., 2025) has been introduced to further refine reasoning processes by optimizing outcome-based rewards, allowing models to explore their reasoning paths. In this paper, we aim to enhance the reasoning capabilities of LLMs in drug discovery.

## 6 CONCLUSION

In this paper, we propose DRUGTRAIL, a reasoning reinforcement learning framework that activates LLMs for interpretable and drug-likeness oriented molecule design. By combining Clinical Chemistry-Informed Reasoning trajectory and a Druggability-Tailored Preference Optimization strategy, DRUGTRAIL achieves efficient and transparent drug discovery. Experiments across multiple tasks confirm its strong generalization, offering a viable path toward AI tools that are both effective and trustworthy for pharmaceutical research.

## ETHICS STATEMENT

We confirm that our work is fully compliant with the ICLR Code of Ethics. This research does not involve any human subjects, sensitive data, or other ethical concerns. All methodologies and applications described align with standard research integrity practices.

## REPRODUCIBILITY STATEMENT

To facilitate reproducibility, we provide a detailed document of the setup in the appendix, covering the overall experimental procedure, dataset acquisition, and specific parameters throughout the process. Furthermore, we validated the robustness of our results in the paper through systematic evaluation and detailed ablation studies. The code is available at https://github.com/Serendipity-r/DrugTrail.

## ACKNOWLEDGEMENTS

This work was supported in part by the National Natural Science Foundation of China No. 62376277, Public Computing Cloud, Renmin University of China, and fund for building world-class universities (disciplines) of Renmin University of China.

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

## A    THE USAGE OF LLMS

In this paper, Large Language Models are utilized for the purpose of polishing the writing.

## B    DETAILS OF TRAJECTORY GENERATION FROM MULTI-OBJECTIVE QUALITY CONTROL

We follow a three-stage sequence for data filtering: Conflict Resolving, Thinking Budget Instruction, and finally Domain Consistency. In the Conflict Resolving stage, we generate ten responses for each sample and perform conflict detection across these responses. For samples that fail conflict detection, we regenerate ten additional responses and reapply the detection process. This procedure is repeated up to three times; samples that still fail after three iterations are discarded. Samples that pass this stage proceed to the Thinking Budget Instruction phase. In this phase, we calculate the mean $\mu$ and standard deviation $\sigma$ of token lengths across all responses for all samples. Any response whose token length exceeds $\mu \pm 3\sigma$ is removed. Finally, the remaining responses advance to the Domain Consistency stage. Samples that pass this final stage are retained for our reasoning dataset; if a sample has more than one valid response at this point, we randomly select one response to include in the dataset.

The detailed statistical information of the data before and after filtering is presented in Tab. 8. Initially, we generate 25,000 reasoning samples from 2,500 original cases. After applying a three-stage data filtering process, the token length of the data is noticeably reduced, and the standard deviation also decreases significantly. This indicates that a substantial portion of excessively long or short samples is removed, resulting in a more complete coverage of the structured reasoning dimensions. We also present the detailed statistics of the selected "Golden" set. These samples are endorsed by human experts as having high content accuracy and logical consistency. In drug design scenarios, the difficulty of ligand generation can vary considerably across different targets; accordingly, the length of reasoning content should adapt to the complexity of the task. Therefore, the samples we select aim to encompass a broad range of token lengths within reasonable limits. The actual statistical results confirm this: while the standard deviation of the "Golden" set is relatively large, the mean length remains within a reasonable range, and all reasoning dimensions are fully represented.

Table 8: Statistical information of the reasoning data before and after filtering. The mean and standard deviation are calculated based on the token lengths.

|  | Number | Mean | Standard deviation | Average number of dimensions |
|---|---|---|---|---|
| Before | 25000 | 2361 | 483 | 4.2 |
| After | 2000 | 1787 | 347 | 5.0 |
| "Golden" set | 15 | 1694 | 835 | 5.0 |

## C    EXPERIMENTAL DETAILS

### C.1    POCKET-LIGAND INTERACTION

#### C.1.1    EVALUATION METRICS

We conduct evaluations on the CrossDocked2020 dataset, following its established split, which yields 100 complexes for testing. Another 100 complexes are randomly selected for validation, leaving 99,900 complexes for training. In the actual SFT training phase, the dataset we construct, which includes reasoning trajectories, contains 2,000 samples.

Following Lin et al. (2024), the evaluation covers four dimensions: substructure, chemical property, geometry, and interaction. However, since our model does not perform direct 3D generation, our evaluation excludes the geometric dimension. The details of each dimension are listed below.

**Substructure** analysis investigates the similarity between generated and reference molecules in terms of atom types, ring types, and functional groups. This evaluation considers how the statistical

distributions of these features correspond across datasets. Two complementary metrics are applied: the Jensen–Shannon divergence (JSD), which measures the distance between probabilistic distributions over molecular substructures, and the mean absolute error (MAE), which reflects differences in average occurrence frequencies (for example, the mean number of carbon atoms per molecule). The atom types in scope include C, N, O, F, P, S, and Cl, while ring types range from three to eight atoms. In addition, functional groups are identified and categorized into 25 classes following the EFG procedure (Salmina et al., 2015).

**Chemical property** evaluation draws upon established cheminformatics metrics, including QED (quantitative estimate of drug-likeness) (Bickerton et al., 2012), SA (synthetic accessibility, normalized to the 0 ĩ range) (Ertl & Schuffenhauer, 2009), and LogP (octanol–water partition coefficient) (Ghose et al., 1999), with optimal values for drug-like candidates typically between 0.4 and 5.6. In addition, the LPSK metric (Lipinski's rule-of-five compliance) (Lipinski, 2004b) calculates the proportion of outputs meeting the specified drug-likeness criteria. Together, these measures quantify attributes such as bioavailability, ease of synthesis, lipophilicity, and compliance with widely accepted medicinal chemistry guidelines.

**Interaction** assessment covers both docking performance and specific interaction patterns. For docking evaluation, three operation modes from AutoDock Vina (Trott & Olson, 2010) are used: Score (direct computation of binding energy for the generated conformation), Min (pose optimization while holding internal coordinates fixed), and Dock (joint optimization of pose and internal structure). For each mode, E denotes the predicted binding energy, and IMP (%) quantifies the fraction of generated molecules with better Vina scores relative to the references. MPBG measures the mean percent binding gap between generated and reference complexes, and LBE expresses the per-atom ligand binding energy contribution. Interaction pattern evaluation employs the PLIP framework (Salmina et al., 2015) to compare the distribution of seven interaction types between generated and reference molecules. Similarity is calculated using JSD and MAE metrics for probabilistic and mean value differences, respectively. Noted that since we does not perform direct 3D generation, we evaluate our method under the docking mode.

### C.1.2 BASELINES

We conduct a comparative analysis between our DRUGTRAIL and both reasoning-enabled and non-reasoning large language models, including DeepSeek-R1 (Guo et al., 2025b), Qwen3-235B-A22B (Yang et al., 2025), QwQ-32B, Qwen3-4B-Thinking (Yang et al., 2025), DeepSeek-V3 (Liu et al., 2024a), Qwen3-235B-A22B-non-thinking (Yang et al., 2025), and Llama-3.1-8B, and further investigate the effectiveness of our proposed DTPO on Qwen3-1.7B, Qwen3-4B and Qwen3-8B.

In addition, we compare our method with several established methods in the field of drug design, including 3DSBDD (Luo et al., 2021), GraphBP (Liu et al.), DiffSBDD (Schneuing et al., 2024), FLAG (Zhang et al., 2023), D3FG (Lin et al., 2023), Pocket2Mol (Peng et al., 2022), and Target-Diff (Guan et al., 2023). As shown in Tab. 9, our performance on the Vina Dock metric, which depends on three-dimensional conformations, is lower than that of models specifically optimized for 3D structural information (e.g., Pocket2Mol and TargetDiff). It is important to note that, unlike these methods, DRUGTRAIL does not directly use 3D structural data as input. Instead, it relies on the amino acid sequence of the protein pocket, leveraging the biomedical knowledge embedded in general-purpose large language models to perform reasoning and molecular generation. Furthermore, it is worth emphasizing that DRUGTRAIL consistently outperforms all baselines on key drug-likeness metrics (QED, SA), indicating that the molecules it generates possess superior synthetic feasibility, drug-like properties, and physicochemical characteristics. This highlights the advantage of our method: achieving comparable binding potential while enhancing the practical prospects of drug development and the transparency of decision-making.

### C.2 MOLECULE EDITING

### C.2.1 TASK SETTINGS

In the molecule editing task, we adopt a zero-shot evaluation setting. Specifically, the model trained on the interaction task using the CCIR and DTPO procedures is directly applied to the molecule editing task without any additional fine-tuning. The prompt used for generation is: *'I am now*

Table 9: Comparison of DRUGTRAIL with other methods on interaction and chemical property of ligands.

| Method | Vina Score ↓ | QED ↑ | LogP | SA ↑ | LPSK ↑ |
|--------|-------------|-------|------|------|--------|
| 3DSBDD | -6.45 | 0.48 | 0.47 | 0.63 | 4.72 |
| GraphBP | -4.57 | 0.44 | 3.29 | 0.64 | 4.73 |
| DiffSBDD | -5.53 | 0.49 | -0.15 | 0.34 | 4.89 |
| FLAG | -3.65 | 0.41 | 0.29 | 0.58 | **4.93** |
| D3FG | -6.78 | 0.49 | 1.56 | 0.66 | 4.84 |
| Pocket2Mol | -7.05 | 0.39 | 2.39 | 0.65 | 4.58 |
| TargetDiff | **-7.41** | 0.49 | 1.13 | 0.60 | 4.57 |
| DRUGTRAIL | -6.82 | **0.57** | 1.71 | **0.72** | 4.81 |

*conducting a molecule editing task. The given molecule is [SMILES]. Please edit it to achieve [task descriptions].*' Following the protocol in Liang et al. (2023), we sample 200 molecules from the ZINC database as the evaluation set.

## C.3 PROTEIN OPTIMIZATION

### C.3.1 EVALUATION METRICS

Following Kirjner et al. (2023), we perform an evaluation on the Green Fluorescent Protein (GFP). We employed three metrics: fitness, diversity, and novelty. It is important to note that higher diversity and novelty do not necessarily indicate superior performance; rather, they offer valuable perspectives on the balance between exploration and exploitation achieved by different approaches. For example, a purely random algorithm would yield the highest possible diversity and novelty (Kirjner et al., 2023).

### C.3.2 TASK SETTINGS

Following Kirjner et al. (2023), we conduct our evaluation on medium and hard difficulty tasks. For the GFP benchmark, task difficulty is determined by two factors: the mutational gap to the optimal fitness and the fitness range of the starting set of sequences. The mutational gap is defined as the minimum number of mutations required to reach the set of optimal proteins, $X^{99th}$, consisting of sequences in the 99th percentile of fitness across the dataset. The fitness range criterion is based on the percentage rank of fitness values in the starting set, with lower ranges presenting greater challenges by requiring methods to identify beneficial mutations from proteins with limited initial functionality. In our study, the medium difficulty setting corresponds to a starting set whose fitness lies between the 20th and 40th percentiles and has a mutational gap of 6. The hard difficulty setting uses a starting set with fitness below the 30th percentile and a mutational gap of 7, making it harder for baseline methods to discover proteins with improved fitness.

## C.4 DTPO TRAINING CONFIGURATIONS

In the DTPO/GRPO training configurations, we use the models (Qwen3-1.7B, Qwen3-4B, and Qwen3-8B) after CCIR/SFT as our base models. For each question, we generate 8 rollouts and restrict the model's maximum response length to 8k tokens. The batch size is set to 64 with one update per training step, which corresponds to an on-policy setting. We adopt a sequence-level mean loss aggregation method and set the KL divergence coefficient $\beta = 0.001$ to maintain training stability. The learning rate is set to $1 \times 10^{-6}$ with a warm-up of 10 steps.

Furthermore, we sample and check the reasoning process during DTPO training. Specifically, we sample a rollout result at each step from step 1 to step 100 of the training process, then perform manual checks and LLM validation. We find that the randomly sampled samples all meet the requirements of consistency and correct, smooth logic. Below, due to the space limits, we randomly display one sampled result.

---

**A sampled rollout result during DTPO training**

```
<Characterization>The protein pocket sequence consists of 60 amino acids. Analysis of physicochemical
  property distribution shows: polar amino acids (e.g., serine, threonine, tyrosine, glutamine)
  account for approximately 50% (positions: Ser13, Thr2,3,5,23,26,34,39,42, Cys14,46, Tyr1
  ,8,19,21,10,19, Glu11,20, Asn42,43, Gln22, Gly25,27,28, Thr41,43,51), forming a high-density polar
  region with extensive hydrogen bonding potential; hydrophobic amino acids (e.g., isoleucine, alanine,
   valine, leucine, methionine, tryptophan) account for approximately 40% (positions: Ile6,17,32, Ala15
  ,24,33,18,29,35,36,40, Leu9, Met26,30, Val4,16,22,44,45,50, Lys31,58,49,54,59, Pro60,52,55, Met30,
  Trp38, Tyr39), forming a hydrophobic patch, especially glycine-rich regions (Gly25-28, Gly27,28) and
  proline turns. Acidic residues (glutamic acid Glu11,20 and aspartic acid Asp12,29,56,57) and basic
  residues (histidine His1,2,47, lysine Lys31,58,49,54,59 and arginine Arg4) are also present,
  suggesting a local electrostatic potential gradient. The overall pocket is amphiphilic, suitable for
  binding ligands with a mixture of polar and nonpolar groups.</Characterization><Stability>Key
  functional residues in the sequence are positioned as follows: hydrophobic/aromatic residue cluster (
  Trp38, Tryptophan on residue 38 can serve as a $\pi$-$\pi$ interaction site; His1,2 and Lys31,47,58
  may participate in hydrogen bonding or $\pi$-stacking); hydrophilic/polar residue group (Ser13, Thr5
  ,23,26,34,42, etc., can act as hydrogen bond acceptors; Tyr1,8,19,21,40,43, Glu11,20, Asp29,56,57,
  Lys31,58,47 may form hydrogen bond donors/acceptors); charged residue regions (negative charges:
  Glu11,20, Asp29,56,57; positive charges: His1,2, Lys31,58,47, Arg4). Linker residues (Gly25,26,27,28,
   Pro55,60) may enhance backbone flexibility. These residues collectively indicate that the ligand
  should be designed to include: hydrophobic scaffolds, hydrogen bonding networks, and charge-
  complementary groups for simultaneous filling and electrostatic balance.</Stability><Guidance>
  Sequence features (such as Cys46, Cys49, His1,2 clustering, and acidic residues concentrated at
  positions 31-47: Lys31-Cys46-Acon39-Cys49-His47) suggest that this pocket may belong to the hydrolase
   or transferase family (Cys/His pattern is commonly found in proteases, such as subtilisins). Prior
  chemical space constraints: The ligand should avoid strong metal coordination (due to infrequent
  metal-binding residues, only Cys46 may act as a nucleophilic site), but can integrate electrophilic
  groups (e.g., aldehyde, ketone) for reversible binding; molecular size is limited to small to medium
  (due to the long sequences Gly25-28 containing glycine forming a flexible region), e.g., weight
  200-400 Da; functional groups need to match the family characteristics, i.e., prefer hydrophobic
  blocks and hydrogen bond acceptors, while excluding large steric groups to prevent clashes.</Guidance
  ><Conservation>No multiple sequence alignment data is available, but based on sequence internal
  pattern conservation, key sites are identified: Asp29 carrying a negative charge (possibly catalytic)
   may be conserved for anion recognition; histidine cluster (His2,47) and Lys58 in the region \"
  HGSNIPE\" (positions 40-46, Glu41) may form a protective function. Ligand design should prioritize
  targeting these sites: anion groups (e.g., carboxylic acid) targeting Lys residues, hydrogen bond
  donors (e.g., primary amine) targeting Asp/Asn/Tyr, while using hydrophobic modules to match the
  polarity transition near positions 40-47.</Conservation><Optimization>Functional group layout: Core
  is a hydrophobic skeleton (e.g., aryl ring or alkyl chain) matching the W38-T39 hydrophobic
  interaction region; periphery arranged polar groups: hydroxyl/carboxyl groups (donor/acceptor)
  targeting Ser/Thr/Tyr clusters (e.g., S13, T26, T34), amino groups (donor) targeting Glu/Asp (
  negative charge), and ketone/carbonyl (acceptor) targeting Lys/His (positive charge); charge
  complementarity design: positive groups (e.g., quaternary ammonium) targeting Glu20/Asp29, negative
  groups (e.g., phosphoric acid) targeting Lys31. Spatial arrangement should avoid functional group
  clustering (e.g., $\pi$ systems near aromatic clusters like Y40, H47), instead have directional
  flexibility (e.g., rotatable bonds) to fit pocket curvature.</Optimization><Answer>Overall
  optimization strategy: Based on pocket's amphiphilic distribution, optimize the ligand to balance
  hydrophobic core (suitable for W38, hydrophobic patch) and polar docking points (utilizing hydrogen
  bond networks driven by S/T polarization and electrostatic complementarity). Strengthen non-covalent
  interactions (e.g., hydrophobic contact points near positions 16-24, and charged interaction points
  near positions 31-47), while designing molecular rigidity-flexibility transitions (e.g., through
  rotatable bonds to adapt to Pro/Gly flexible regions), ensuring binding affinity and specificity. \\
  boxed{COc1ncc(NCC(=O)NO)c(CCN2CCN(CC(=O)O)CC2)n1}</Answer>
```

# D  DETAILED RELATED WORK

In this part, we present detailed related work on the application of Large Language Models (LLMs) to drug discovery as an extention of § 5. Specifically, existing approaches to adapt LLMs for scientific applications can be roughly grouped into three categories. First, a typical line of work trains models from scratch for specific domainssuch as sequentialized proteins (Chen et al., 2024; Hayes et al., 2025; Zhu et al., 2025), DNA (Nguyen et al., 2024), small molecules (Liu et al., 2023), or scientific corpora (Luo et al., 2022; Taylor et al., 2022). They achieve accuracy within their respective domains but require massive amounts of in-domain data and lack flexibility for cross-domain understanding. Second, several studies have explored fine-tuning general-purpose LLMs on molecular (Chaves et al., 2024) or protein data (Lv et al., 2025). However, they still requires millions of training samples, resulting in high computational and resource costs. The third line of approaches integrate pretrained scientific encoders into LLMs through lightweight adapters, which leverages multi-modal information but faces challenges in aligning heterogeneous representations (Liang et al., 2023; Luo et al., 2023; Li et al., 2025). More recently, unified frameworks such as NatureLM (Xia et al., 2025) attempt to provide broad scientific coverage by pretraining on a sufficiently large mixed-domain corpora. While such models show strong potential, they remain heavily dependent on large-scale data and primarily focus on generating final predictions rather than offering transparent intermediate reasoning. Overall, despite notable progress, current LLM-based methods for drug discovery are still constrained

by costly data requirements and a lack of explicit reasoning processes, underscoring the need for approaches that combine interpretability with efficiency.

# E  DETAILED PROMPTS AND GENERATED EXAMPLES

---

### Prompt for getting structured reasoning dimension

**[Pocket-Ligand Interaction]**

I am currently working on a pocket−ligand interaction task, in which the goal is: given the sequence of a protein pocket, to design candidate drug molecules capable of binding to it and producing meaningful interactions. In this design process, suppose you are acting as a pharmacologist: What key and broadly applicable aspects should you take into consideration? Please outline several critical and general factors.

**[Protein Editing]**

I am currently working on a protein optimization task, where the objective is: given the sequence of a protein and a target property to be improved, to design new protein sequences that meet the desired requirements for that property. In this design process, suppose you are acting as a biologist: What key and broadly applicable aspects should you take into consideration? Please identify several critical and general factors.

---

### Prompt for getting structured reasoning trajectory

**[Pocket-Ligand Interaction]**

The goal is to perform a pocket−ligand interaction task. I will give you the sequence of the protein pocket, and you need to generate the corresponding ligand molecule SMILES that can bind to this pocket. I will also provide the ligand molecule SMILES that can bind to it, which is the correct answer. Your task is: Based on the given protein pocket sequence and the sequence information that can be directly calculated, use the following five fixed analytical dimensions:
In−depth physicochemical profiling, Maintenance of core structural and functional integrity, Prior knowledge and chemical / structural space guidance, Conservation analysis and identification of critical sites, Optimization and multi−attribute balancing, to deduce a ligand that can bind to the given pocket −that is, the ligand molecule SMILES I provided.
During the reasoning process:
Do not use, directly or indirectly, any information from the ligand molecule SMILES I provided. Strictly follow this rule.
Do not use or guess any experimental data or geometric/physical quantities.
The amino acid position numbers in the given protein pocket sequence start from 1 −strictly follow this rule.
Input:
Protein pocket sequence: {protein}
Ligand molecule SMILES: {ligand}
Output requirements:
Output five reasoning sections in the order 1−5. Do not directly or indirectly use any information from the ligand molecule SMILES I provided.
At the end, output a general overall optimization idea and then give the answer.

**[Protein Editing]**

The goal is to perform a protein optimization task. I will give you the protein sequence to be optimized, and you need to optimize the green fluorescence properties of the protein by editing the amino acid types in the sequence. I will also provide an optimized protein sequence (with stronger green fluorescence than the target sequence), which is the correct answer. Your task is: Based on the given protein sequence and the sequence information that can be directly calculated, use the following five fixed analytical dimensions:
In−depth physicochemical profiling, Maintenance of core structural and functional integrity, Prior knowledge and chemical / structural space guidance, Conservation analysis and identification of critical sites, Optimization and multi−attribute balancing, to derive a more optimal protein sequence, based on the goal of enhancing fluorescence properties −that is, the optimized protein sequence I provided.
During the reasoning process:
Do not use, directly or indirectly, any information from the optimized protein sequence I provided. Strictly follow this rule.
Do not use or guess any experimental data or geometric/physical quantities.
The amino acid position numbers in the given protein pocket sequence start from 1 −strictly follow this rule.
Input:
Protein sequence to be optimized: {protein}
Optimized protein sequence: {target protein}
Output requirements:
Output five reasoning sections in the order 1−5. Do not directly or indirectly use any information from the optimized protein sequence I provided.
At the end, output a general overall optimization idea and then give the answer.

---

---

**An example of explainable drug discovery process**

**[Question]**

I am currently conducting a pocket–ligand interaction task. The given pocket is {'A':'IGSFGTVYKVAVKLIVTQWCNFGDDLEGSSLHKINRGHMGAFMGLFGLLKIDS'}. Please design a ligand molecule capable of binding to it.

**[Answer]**

<Characterization>The pocket sequence contains a high proportion and uniform distribution of hydrophobic residues (I, L, V, F, M, A, G), suggesting the presence of a sizable hydrophobic cavity. Aromatic residues (F, Y, W) are located at multiple positions and may form $\pi$–$\pi$ stacking with aromatic rings or hydrophobic fragments in the ligand. In designing the ligand, incorporating a stable hydrophobic aromatic core is recommended to provide both stacking interactions and a spatial scaffold for orientation within the pocket.</Characterization><Stability>Multiple polar residues (S, T, N, Q, C) and charged residues (K, R, D, E, H) embedded in a hydrophobic background indicate discrete polar "hot spots." A ligand could make use of these by introducing a polar functional group (e.g., carbonyl, amide, hydroxyl, or heteroaromatic nitrogen) at one end or a key position on the aromatic scaffold to act as a hydrogen–bond donor/acceptor, thus enabling directional H–bond contacts with those functional residues.</Stability><Guidance>Motif features suggest similarity to certain kinase or nucleotide–binding sites, which typically accommodate a planar aromatic or heteroaromatic core mimicking ATP or related substrates. For such sites, placing a stable aromatic or heteroaromatic scaffold at the center of the ligand is beneficial, occupying the binding plane while offering edge positions for substituents to reach peripheral hydrophobic or polar regions. Incorporating one or more heteroatoms into the scaffold can further engage conserved polar residues through hydrogen bonding.</Guidance><Conservation>Likely conserved regions may include acidic residues (D/E) and glycine–rich flexible motifs–often critical for anchoring small polar moieties. To address these, specific positions on the ligand scaffold could be designed to carry H–bond donor or acceptor groups such as aromatic nitrogens, amide carbonyls, or N–H donors, enabling compatibility with conserved residue interaction patterns and improving binding affinity.</Conservation><Optimization>Overall, a ligand layout could combine: A strong hydrophobic/aromatic core for stacking in the cavity. One or more strategically positioned polar anchors to engage hydrogen–bond sites or charged residues. A moderately bulky hydrophobic substituent at one terminus or side chain (e.g., fluoroalkyl, cycloalkyl, substituted aryl) to fit into small hydrophobic subpockets.</Optimization><Answer>Based on the five–dimensional analysis, I have designed a molecule, results in \boxed{CC1=C(C=C(C=C1)C(=O)NC2=CC=CC(=C2)C(F)(F)F)C3=CC(=NC=C3)N4CCOCC4}.</Answer>

---

