# OpenReview forum: "DrugTrail: Interpretable Drug Discovery via Structured Reasoning and Druggability‑Tailored Preference Optimization"
_ICLR.cc/2026/Conference — ICLR 2026 Poster_

### Official Review · Reviewer_6Cmn · 2025-10-28

**Soundness:** 4
**Presentation:** 4
**Contribution:** 4
**Rating:** 8
**Confidence:** 3

**Summary:**

The authors present DrugTrail, an LLM-based method for computational drug discovery. DrugTrail is a novel approach to an ongoing challenge in drug discovery: the de novo generation of drug-like molecules. Through a RL-based process to optimize several metrics (beyond binding affinity), DrugTrail can effectively generate small molecules/ ligands for binding proteins. This is an interesting, interpretable approach which has several steps to incorporate and align with biochemical domain knowledge.

Overall, this is a nice paper, and I will recommend an acceptance. However, the authors should provide better explanations of some of the biochemical jargon (see below).

**Strengths:**

**Strong points:**

- Unique from other approaches
- Great use of relevant scientific and medicinal information for model context.
- The approach for tokenization is very sensible for the application.
- The authors clearly thought-out every step of the pipeline. I appreciate that every step is justified appropriately.
- Great figures.
- The claims follow the results.
- The background information motivates the need for DrugTrail well.

**Weaknesses:**

**Weak points:**
- My main criticism is that some of the biochemical jargon needs to be better or earlier explained. I know this is difficult with limited space, but it is important for understanding the paper, especially for a computational conference.
- The explanation of biochemically relevant acronyms, such as LPSK and QED, are in the supplementary, but this makes it difficult to understand what these are when they come up earlier in the main text and corresponding figures.

**Questions:**

- Please go through all the acronyms and ensure they are defined before being used.
- In your paper, better define biochemical jargon. A few examples are included below, but this is not an exhaustive list.
    - "ligand" - this is also used interchangeably with "molecule" or "small molecule" without an explanation that they are synonymous
    - "druggable"
    - "binding pocket"
    - "canonical SMILES"
    - "backbone"
    - "docking"

---

> ### Author Response · Authors · 2025-11-23
>
> ***Weakness 1 & Question 2:***
>
> >W1: My main criticism is that some of the biochemical jargon needs to be better or earlier explained. I know this is difficult with limited space, but it is important for understanding the paper, especially for a computational conference.
>
> >Q2: In your paper, better define biochemical jargon. A few examples are included below, but this is not an exhaustive list.
>
> We sincerely appreciate your recognition of our work and your valuable feedback. We have carefully supplemented and revised the definitions of the relevant terms in the manuscript.
>
> ***Weakness 2 & Question 1:***
>
> >W2: The explanation of biochemically relevant acronyms, such as LPSK and QED, are in the supplementary, but this makes it difficult to understand what these are when they come up earlier in the main text and corresponding figures.
>
> >Q1: Please go through all the acronyms and ensure they are defined before being used.
>
> Thank you for your constructive comments. This is indeed an important point that warrants attention. We have carefully provided explanations and definitions for the relevant terms in the manuscript.

---

> ### Author Response · Authors · 2025-11-27
>
> Dear Reviewer  6Cmn,
>
> We are deeply grateful for your thorough review and for the many positive remarks on our work, particularly your recognition of DrugTrail’s uniqueness, the thoughtful design of its workflow, and the clarity of our figures and claims. Your encouraging words mean a great deal to us.
>
> In response to your suggestions, we have made improvements to address the points you raised:
>
> - We expanded and revised explanations of biochemical terms and **acronyms**, ensuring they are defined before use in the main text rather than only in the supplementary materials.
>
> - We clarified the **meanings of terms** such as ligand, binding pocket, druggable, and canonical SMILES, to make them more accessible for readers.
>
> We would like to thank you again here for your recognition and encouragement of our work, and for the constructive suggestions you made that greatly improved the writing quality of our work.
>
> Best regards,
>
> Authors

---

### Official Review · Reviewer_ru54 · 2025-10-28

**Soundness:** 3
**Presentation:** 3
**Contribution:** 3
**Rating:** 6
**Confidence:** 3

**Summary:**

DRUGTRAIL is an LLM-based framework for interpretable drug discovery that combines structured reasoning and reinforcement learning. It applies a Druggability-Tailored Preference Optimization (DTPO) scheme to balance binding affinity and drug-likeness. Trained mainly on CrossDocked2020, it outperforms baseline LLMs in docking and property metrics.

**Strengths:**

1. By enforcing explicit reasoning trajectories, DRUGTRAIL makes the model’s decisions transparent and closer to human medicinal-chemistry logic.
2. Demonstrates consistent improvements across interaction, chemical, and structural metrics, and shows transferability to both small- and large-molecule tasks.
3. The creation of a reasoning dataset with conflict resolution, domain consistency checks, and a thinking budget is novel and carefully designed.

**Weaknesses:**

1. Do the authors provide a rationale for designing the reasoning format with tags such as Characterization, Stability, etc? It is unclear why these particular dimensions were chosen.
2. Only general LLMs (eg. Qwen) are used for comparison. There is no evaluation against established drug-design systems.

**Questions:**

see weakness

---

> ### Author Response · Authors · 2025-11-23
>
> ***Weakness 1:***
>
> >Do the authors provide a rationale for designing the reasoning format with tags such as Characterization, Stability, etc? It is unclear why these particular dimensions were chosen.
>
> We sincerely appreciate your valuable feedback.
>
> Regarding the selection of the reasoning dimensions, **our method is based on a systematic emulation of the core cognitive patterns employed by medicinal chemists in ligand design**. Specifically, we first pose a question to a large language model (as detailed in **Appendix G.1** of the paper): when designing ligands based on a protein pocket, what essential and broadly applicable factors should a pharmacologist consider? **Based on the model’s responses, we distill five analytical perspectives that are both critical and widely relevant in drug design contexts**, namely: In-depth physicochemical profiling, Maintenance of core structural and functional integrity, Prior knowledge and chemical/structural space guidance, Conservation analysis and identification of critical sites, and Optimization and multi-attribute balancing. These five dimensions together form the framework of the structured reasoning process.
>
> In the training process, **we map each of the five reasoning dimensions to structured labels in order to guide the model toward generating reasoning trajectories consistent with domain-specific cognitive logic**. In practice, we find that using the full names of these dimensions directly as labels posed difficulties in training due to excessive label length. Therefore, **we summarize each reasoning dimension into a single-word label**, namely: Characterization, Stability, Guidance, Conservation, and Optimization.
>
> ***Weakness 2:***
>
> >Only general LLMs (eg. Qwen) are used for comparison. There is no evaluation against established drug-design systems.
>
> Thank you for your valuable suggestions.
>
> **In Section E.1.2 of the Appendix, we have supplemented the manuscript with a performance comparison against other drug design methods, as presented in Table 9**. It is worth noting that, unlike other methods, our method does not directly use 3D structural information as input. Instead, it relies on the amino acid sequence of the protein pocket, leveraging the biomedical knowledge embedded in general-purpose large language models to perform reasoning and molecular generation.
>
> Consequently, in terms of Vina Dock, an evaluation metric that depends on 3D conformations, our method performs at a mid-range level compared with 3D-specific generative models, and there remains a slight performance gap between our method and State-Of-The-Art models (e.g., Pocket2Mol [1], TargetDiff [2]). However, it is important to emphasize that our method consistently outperforms all baselines on drug-likeness metrics (QED, SA), indicating that its generated molecules exhibit superior synthetic feasibility, drug-likeness, and physicochemical profiles. This result highlights the strength of our method: **maintaining comparable binding potential to specialized models while improving the practical prospects for drug development and the transparency of decision-making**.
>
> Once again, thank you for your valuable comments.
>
> [1] Peng X, Luo S, Guan J, et al. Pocket2mol: Efficient molecular sampling based on 3d protein pockets[C]//International conference on machine learning. PMLR, 2022: 17644-17655.
>
> [2] Guan J, Qian W W, Peng X, et al. 3D Equivariant Diffusion for Target-Aware Molecule Generation and Affinity Prediction[C]//The Eleventh International Conference on Learning Representations.

---

> ### Author Response · Authors · 2025-11-27
>
> Dear Reviewer ru54,
>
> Thank you sincerely for your thoughtful review and for recognising the strengths of our work, especially the focus on novelty, logical consistency, and transferability across different tasks. Your acknowledgement of these aspects has been very encouraging for us.
>
> According to your feedback, we have made concrete clarifications, supplementary experiments and manuscript revisions to address the raised points.
>
> - We provided a detailed clarification regarding **the selection for reasoning dimensions**.
>
> - We supplemented the manuscript with a performance **comparison against other drug design methods** in Section E.1.2 and Table 9.
>
> We hope these additions and explanations help to resolve the uncertainties you mentioned. If possible, we would greatly value any further comments you would like to share. Your review has greatly improved the completeness of our work.
>
> Thank you again for your engagement and constructive feedback. We truly look forward to hearing from you.
>
> Best regards,
>
> Authors

---

### Official Review · Reviewer_Pp95 · 2025-10-31

**Soundness:** 3
**Presentation:** 3
**Contribution:** 3
**Rating:** 6
**Confidence:** 4

**Summary:**

This paper presents DRUGTRAIL, an LLM framework for explainable drug discovery. It first uses Clinical Chemistry-Informed Reasoning (CCIR) to fine-tune a model on structured reasoning, then applies Druggability-Tailored Preference Optimization (DTPO) to optimize for multi-component rewards (ligand similarity, drug-likeness, and reasoning format) beyond simple binding affinity.

**Strengths:**

The work commendably tackles the critical challenges of interpretability and multi-objective optimization in drug discovery. Its core strength is the DTPO reward function, which explicitly moves beyond optimizing affinity scores by balancing structural similarity, rule-based drug-likeness, and reasoning coherence, addressing a key limitation of prior methods.

**Weaknesses:**

The "Reasoning Quality Reward" is purely syntactic, rewarding formatting tags rather than semantic accuracy, which undermines the interpretability claim. Furthermore, the evaluation risks circularity, as the model is rewarded for similarity to a Vina-filtered dataset and then primarily evaluated with Vina. Finally, it fails to resolve the 1D/3D contradiction: the model uses 1D sequences but makes 3D-dependent inferences (e.g., π-π stacking) and relies on 3D docking for evaluation.

**Questions:**

1.There is an inconsistency in the description of the reasoning dimensions. Section 2.1.1 explicitly lists "five core reasoning dimensions". However, Section 2.1.3 states the SFT data conforms to "six predefined reasoning dimensions", apparently counting the final <Answer> block as the sixth. This is confusing.

2.The SFT dataset generation (2.1.2) relies on several LLM-based filtering steps, such as Conflict Resolving and Domain Consistency which introduce unquantified biases. The "Domain Consistency" check, in particular, relies on a "small set of 'golden' reasoning trajectories" whose size and diversity are not specified.

3.By pre-filtering the reference dataset with AutoDock Vina and then rewarding Tanimoto similarity, the model is effectively trained to optimize the Vina score, which is also the primary evaluation metric (Table 1). This risks overfitting to the Vina function.

4.The "Reasoning Quality Reward" merely checks for the syntactic presence of formatting tags, not the semantic quality or logical accuracy of the reasoning content. Sometimes the model is not penalized for generating nonsensical reasoning as long as the format is correct.

5.The paper claims its 1D SMILES generation method "excludes the geometric dimension". However, pocket-ligand binding is inherently a 3D problem. Furthermore, the model's inference (e.g.,π-π stacking) and key evaluation (Vina docking score) heavily rely on 3D geometry. It’s better that explain how the model learns this implicit 3D perception solely from 1D sequences.

---

> ### Author Response · Authors · 2025-11-23
>
> ***Weakness 1 & Question4:***
>
> >W1: The "Reasoning Quality Reward" is purely syntactic, rewarding formatting tags rather than semantic accuracy, which undermines the interpretability claim.
>
> >Q4: The "Reasoning Quality Reward" merely checks for the syntactic presence of formatting tags, not the semantic quality or logical accuracy of the reasoning content. Sometimes the model is not penalized for generating nonsensical reasoning as long as the format is correct.
>
> We sincerely appreciate your valuable comments and are very pleased to engage in a discussion on this matter.
>
> Our reinforcement learning (RL) process adopts the Group Relative Policy Optimization (GRPO) framework, which is consistent with the paradigm employed in **Deepseek-r1**[1]. This framework separates the functions of different reward components, where format-related rewards guide the model’s thinking patterns, while outcome-level rewards shape the semantic correctness and domain validity of the final solution. In mathematical tasks, this approach yields a chain-of-thought reasoning process characterized by strong consistency and rigorous logical thinking, even without explicity process-level supervision. Our drug design task shares similar characteristics with Deepseek-r1. **In both settings, obtaining a correct final result from an illogical or nonsensical intermediate reasoning process is generally unlikely**. Consequently, the outcome-level reward implicitly enforces semantic validity, because syntactically well-organized but semantically incoherent reasoning would hardly lead to a successful outcome and therefore cannot be reinforced. **This task difficulty reduces the risk of simply hacking format-oriented rewards**.
>
> In addition, prior to RL optimization, we introduce a supervised fine-tuning (SFT) stage to enhance the reasoning capabilities. In this stage, a three-step data filtering and selection process is applied to **obtain a set of high-quality intermediate structured reasoning samples**, which are then incorporated into the model. **This served as an initial guidance for strengthening the model’s intermediate reasoning capabilities**.
>
> In practice, our purpose in setting the Reasoning Quality Reward is to **encourage the model, particularly during the early phase of RL, to adopt structured thinking patterns from multiple perspectives**. Working together with other outcome rewards, this also stimulates the model’s biological capability. As shown in **Figure 4**, the Reasoning Quality Reward curve rises rapidly at the early stage of training, while the other rewards increase gradually as training progresses.
>
> Moreover, we **sample and examine rollout results from the RL stage** and find that the model is able to generate logically coherent and meaningful intermediate reasoning processes, which are presented in **Section E.4** of the manuscript.
>
> Once again, thank you for your valuable comments.
>
> ***Weakness 2 & Question3:***
>
> >W2: Furthermore, the evaluation risks circularity, as the model is rewarded for similarity to a Vina-filtered dataset and then primarily evaluated with Vina.
>
> >Q3: By pre-filtering the reference dataset with AutoDock Vina and then rewarding Tanimoto similarity, the model is effectively trained to optimize the Vina score, which is also the primary evaluation metric (Table 1). This risks overfitting to the Vina function.
>
> Thank you for your valuable feedback. **In Section 4.1 of the manuscript, we conduct supplementary experiments to better validate the performance of our model**.
>
> It is worth noting that **our method is intended to provide a general framework aimed at generating high-quality, high-affinity ligands, rather than focusing on the Vina tool itself**. To this end, in the supplementary experiments we explore multiple screening and ranking strategies when constructing the reference compound set. **In addition to AutoDock Vina, we also employ Boltz2 [2] (a highly efficient affinity prediction tool with accuracy approaching that of free energy perturbation methods)  and pIC50 labels experimentally verified in wet-lab settings from the SIU dataset [3] to rank candidate molecules**.
>
> Accordingly, **at the evaluation stage we report both the Vina Score and Boltz2-predicted affinity** as performance metrics. As shown in **Table 7**, across all six combinations of “screening strategy × evaluation metric”, our method consistently demonstrated competitive performance.
>
> Once again, we are grateful for your insightful comments, which have helped us present a more rigorous validation of our method.

---

> ### Author Response · Authors · 2025-11-23
>
> ***Weakness 3 & Question5:***
>
> >W3: Finally, it fails to resolve the 1D/3D contradiction: the model uses 1D sequences but makes 3D-dependent inferences (e.g., π-π stacking) and relies on 3D docking for evaluation.
>
> >Q5: The paper claims its 1D SMILES generation method "excludes the geometric dimension". However, pocket-ligand binding is inherently a 3D problem. Furthermore, the model's inference (e.g.,π-π stacking) and key evaluation (Vina docking score) heavily rely on 3D geometry. It’s better that explain how the model learns this implicit 3D perception solely from 1D sequences.
>
> Thank you for your valuable comments. This is indeed a nice question. Fundamentally, geometric analysis is important for understanding ligand–binding interactions within target pockets. In our study, we attempt to **implicitly incorporate geometric analysis through three dimensions in the structured reasoning process**: Maintenance of core structural and functional integrity, Prior knowledge and chemical/structural space guidance, and Conservation analysis and identification of critical sites.
>
> The first dimension focuses on analyzing, **from the perspective of maintaining core structural and functional integrity**, which chemical scaffolds, functional groups, or spatial conformations are essential for binding ( e.g. forming specific hydrogen bonds, salt bridges, or π–π stacking). During ligand design, it is crucial to preserve the position and orientation of functional groups that interact with these residues.
> The second dimension aims to provide prior knowledge and guidance within chemical and structural space, including **analyzing the protein family of the target and leveraging established inhibitors or binding patterns from related family members to guide the design process**.
> The third dimension involves **identifying conserved regions and key amino acid residues through homology-based structural alignment and conservation analysis**. Such conserved sites often play a decisive role in determining ligand binding location and mode.
>
> **As discussed in Section 4.1 of the manuscript and presented in Table 6, we carry out supplementary experiments in which these three dimensions are removed from the structured reasoning process. Retraining and evaluating the model under these conditions resulted in a notable decline in performance. This finding demonstrates that geometry-related analysis provides positive informational gains for drug design**. It also aligns with our objective of activating relevant knowledge already embedded in large language models to support drug discovery tasks.
>
> We acknowledge that direct 3D coordinate information is not included in the current study, and this indeed remains an important direction for future research.
>
> ***Question 1:***
>
> >There is an inconsistency in the description of the reasoning dimensions. Section 2.1.1 explicitly lists "five core reasoning dimensions". However, Section 2.1.3 states the SFT data conforms to "six predefined reasoning dimensions", apparently counting the final \<Answer\>…\</Answer\> block as the sixth. This is confusing.
>
> Thank you for your suggestions, and we have carefully revised the relevant wording and statements in the manuscript. Our SFT data include the five reasoning dimensions listed in Section 2.1.1, as well as the final answer ( \<Answer\>…\</Answer\> ). We sincerely apologize for the confusion.
>
> ***Question2:***
>
> >The SFT dataset generation (2.1.2) relies on several LLM-based filtering steps, such as Conflict Resolving and Domain Consistency which introduce unquantified biases. The "Domain Consistency" check, in particular, relies on a "small set of 'golden' reasoning trajectories" whose size and diversity are not specified.
>
> We are grateful for the valuable comments. The quality of reasoning data is indeed critical to the performance of the model.
>
> However, manual data annotation and quality evaluation are both challenging and costly. **Considering that large language models have, to some extent, learned human preferences during their large-scale pre-training**, we choose them as an effective alternative for data filtering and selection.
>
> **In Appendix D of the manuscript, we have provided additional statistical information before and after data filtering**. As shown in **Table 8**, data filtering resulted in a notable reduction in both the mean and variance of token counts in the structured reasoning process. At the same time, the average number of reasoning dimensions covered per text increased significantly. Furthermore, we have included details of the Domain Consistency step to illustrate that the selected “Golden” set possesses both high data quality and diversity.

---

> ### Author Response · Authors · 2025-11-23
> **References**
>
> [1] Guo D, Yang D, Zhang H, et al. Deepseek-r1 incentivizes reasoning in llms through reinforcement learning[J]. Nature, 2025, 645(8081): 633-638.
>
> [2] Passaro S, Corso G, Wohlwend J, et al. Boltz-2: Towards accurate and efficient binding affinity prediction[J]. BioRxiv, 2025.
>
> [3] Huang Y, Gao B, Jia Y, et al. SIU: A Million-Scale Structural Small Molecule-Protein Interaction Dataset for Unbiased Bioactivity Prediction[J]. arXiv preprint arXiv:2406.08961, 2024.

---

> ### Author Response · Authors · 2025-11-27
>
> Dear Reviewer Pp95,
>
> We truly appreciate the time and effort you have devoted to reviewing our paper and the encouraging recognition of its strengths,  especially the focus on interpretability, multi-objective optimization, and the DTPO reward function. Your positive comments are very encouraging to us.
>
> Since receiving your feedback, we have undertaken targeted manuscript revisions and additional experiments to address the raised concerns.
>
> - We explained the design and role of the **Reasoning Quality Reward** and added details on how it supports coherent, meaningful reasoning content (Section E.4 & Figure 4).
>
> - We conducted supplementary experiments with additional docking methods, including Boltz2 and SIU, to reduce **evaluation bias** (Section4.1 & Table 7).
>
> - We included additional geometric analysis (Section 4.1) and ran supplementary control experiments (Table 6). Results show that **geometry-related reasoning** gives notable performance gains.
>
> - We clarified **dataset** generation and addressed the **description inconsistency** in the reasoning dimensions, adding diversity and quality details in Appendix D and Table 8.
>
> We hope these updates help to resolve your concerns, and we would be very grateful for any additional suggestions you might have. Your valuable comments have greatly help us to strengthen the work.
>
> Thank you again for your constructive input and support. We sincerely look forward to your reply.
>
> Best regards,
>
> Authors

---

### Official Review · Reviewer_aDhV · 2025-10-31

**Soundness:** 3
**Presentation:** 3
**Contribution:** 2
**Rating:** 6
**Confidence:** 4

**Summary:**

Summary:
The paper introduces DRUGTRIAL, an LLM framework for explainable drug discovery. It try to addresses the “black-box” nature of AI methods by introducing structured reasoning traces to explain how and why behind its conlcusion. The work also uses DTPO strategy to optimize not only for binding affinity but also balance multiple essential factors.

**Strengths:**

Pros:
- The paper introduces DRUGTRIAL, an LLM framework for explainable drug discovery.
- Extensive experiments demonstrate the effectiveness of our approach and its generalizability to a wider range of biomolecular optimization domains.

**Weaknesses:**

Cons:
- 3D structural analysis doesn’t seem to taken into account, which is the core for small molecule drug design from medicinal chemist viewpoint.
- Explainability is a big claim; this method is clearly not explainable but only can provide interpretable insights. Phrasing and statements regarding this are encouraged to modified.
- Vina is an old approach that is not considered very accurate. More advanced binidng affinity prediction or docking methods like Boltz2, and PSICHIC, can be considered.
* hyperparameters for multiple reward functions may be difficult to tune.

**Questions:**

See Weaknesses

---

> ### Author Response · Authors · 2025-11-23
>
> ***Weakness 1:***
>
> >3D structural analysis doesn’t seem to taken into account, which is the core for small molecule drug design from medicinal chemist viewpoint.
>
> We sincerely appreciate your valuable feedback. Geometric analysis is indeed important in drug design. In fact, our study attempts to **perform implicit geometric analysis through the three dimensions of the structured reasoning process**, namely: Maintenance of core structural and functional integrity, Prior knowledge and chemical/structural space guidance, and Conservation analysis and identification of critical sites.
>
> The first dimension aims to examine, from the perspective of **core structural and functional integrity**, which chemical scaffolds, functional groups, or spatial conformations are essential for binding—such as the formation of specific hydrogen bonds, salt bridges, or π–π stacking. In ligand design, it is important to retain the position and orientation of functional groups that interact with these residues.
> The second dimension focuses on providing prior knowledge and guidance based on chemical and structural space. This involves **analyzing the target’s protein family and leveraging known inhibitors or binding patterns from related family members to inform the design process**.
> The third dimension identifies **conserved regions and key amino acid residues through homology-based structural alignment and conservation analysis**. These conserved sites typically determine the binding position and mode of ligands.
>
> **In Section 4.1 of our paper, we also conduct supplementary experiments. As shown in Table 6, when these three dimensions are excluded from the structured reasoning process and the model is retrained and evaluated, its performance declines significantly. This finding demonstrates that geometry-related analysis provides positive informational gains in drug design**. Furthermore, it aligns with our goal of activating relevant knowledge already embedded in large language models to support drug design tasks.
>
> We acknowledge that direct 3D coordinate information is not incorporated into the current workflow, which indeed represents an important and promising avenue for our future research.
>
> ***Weakness 2:***
>
> >Explainability is a big claim; this method is clearly not explainable but only can provide interpretable insights. Phrasing and statements regarding this are encouraged to modified.
>
> We sincerely appreciate you for pointing this out. We have carefully revised the relevant wording and statements in the manuscript, and have accordingly modified the title.
>
> ***Weakness 3:***
>
> >Vina is an old approach that is not considered very accurate. More advanced binidng affinity prediction or docking methods like Boltz2, and PSICHIC, can be considered.
>
> We are grateful for your constructive suggestions. **In Section 4.1 of the manuscript, we have conducted supplementary experiments**. In the experiments, we employ two methods, **Vina and Boltz2 [1]**, to screen and rank the bioactive compounds, and the model performance is evaluated separately under both metrics. As shown in **Table 7**, our method achieves consistently strong performance across all the four experimental settings.
>
> ***Weakness 4:***
>
> >hyperparameters for multiple reward functions may be difficult to tune.
>
> Thank you for your comments. As noted in [2], training multiple reward functions indeed requires balancing the weights among them. In **Section 4.1** of our manuscript, we investigate and discuss this issue, and obtain several notable findings.
>
> As illustrated in Figure 2(c), we conduct experiments to examine the specific role of each of the three reward functions. Our results show that removing the reasoning quality reward leads to a significant drop in model performance, indicating that a structured biological reasoning pattern is critical for reinforcement learning. In addition, the ligand-based reward plays a key role in enhancing interaction awareness, while the rule-based reward primarily improves drug-likeness. Effective drug discovery requires achieving a balance between binding affinity and drug-likeness.
>
> Subsequently, as shown in Figure 2(a), we carry out experiments to explore how such a balance could be achieved. We find that an excessive emphasis on the reasoning quality reward can limit the model’s perception of specific biological capabilities. Ultimately, across six sets of parameter configurations, we obtain results that we consider both balanced and satisfactory.
>
> [1] Passaro S, Corso G, Wohlwend J, et al. Boltz-2: Towards accurate and efficient binding affinity prediction[J]. BioRxiv, 2025.
>
> [2] Peng H, Qi Y, Wang X, et al. Agentic reward modeling: Integrating human preferences with verifiable correctness signals for reliable reward systems[J]. arXiv preprint arXiv:2502.19328, 2025.

---

> ### Author Response · Authors · 2025-11-27
>
> Dear Reviewer aDhV,
>
> Thank you very much for your careful review and for recognizing the contributions of our work, especially the introduction of the framework and its potential generalizability. Your positive comments are greatly encouraging to us.
>
> Following your feedback, we have carefully added new experiments, clarifications, and manuscript revisions to address each of the concerns:
>
> - **3D structural analysis**: We included additional geometric analysis (Section 4.1) and ran supplementary  experiments (Table 6). Results show that geometry-related reasoning gives clear performance gains.
>
> - **Explainability claim**: We revised the words and adjusted the title accordingly.
>
> - **Docking methods**: We added experiments with Boltz2 (Table 7) , showing consistent improvements across metrics.
>
> - **Multiple reward functions**: We analyzed ablation studies (Figure 2) to further clarify our findings.
>
> We hope these additions address your questions and would greatly appreciate any further comments or discussion from you. Your guidance has already help us to improve our work.
>
> Thank you again for your time and support, and we look forward to hearing from you.
>
> Best regards,
>
> Authors

---

### Author Response · Authors · 2025-11-23
**General Response**

Dear ACs and reviewers,

We sincerely thank you for your time, effort, and constructive feedback on our work. This work mainly focuses on interpretable drug discovery via structured reasoning and druggability‑tailored preference optimization. We are glad that the reviewers recognized the contributions of our paper, which we briefly summarize as follows.

 - **Novelty:** "tackle the critical challenges of interpretability and multi-objective optimization in drug discovery and address a key limitation of prior methods" (Pp95) "The creation of a reasoning dataset is novel and carefully designed." (ru54) "Unique from other approaches." (6Cmn)

- **Presentation:** "The authors clearly thought-out every step of the pipeline. I appreciate that every step is justified appropriately." "Great figures." "The background information motivates the need for DrugTrail well." (6Cmn)

- **Experiments:** "Extensive experiments demonstrate the effectiveness of the approach and its generalizability to a wider range of biomolecular optimization domains." (aDhV) "shows transferability to both small- and large-molecule tasks" (ru54) "The claims follow the results." (6Cmn)

We appreciate the insightful comments and suggestions from the reviewers, and provide additional experimental results, clarifications, and manuscript revisions during the rebuttal phase. Below we summarize the main updates.

- **Additional Results:** We conducted an ablation study by removing the geometry-related reasoning dimensions (Sec.4.1, Tab.6) and confirmed their significant contribution to overall performance (aDhV Pp95). We adopted Boltz2 and experimentally measured pIC50 data as alternatives or supplements to AutoDock Vina for compound screening and evaluation (Tab.7), validating the stability and generalization ability of the model across multiple screening–evaluation combinations (aDhV Pp95). We added performance comparisons with established drug design methods (e.g.Pocket2Mol, TargetDiff) (Appendix E.1.2, Tab.9) (ru54).

- **Clarifications:** Following the reviewer’s suggestion, we revised “explainable” to “interpretable” and updated the paper title accordingly (aDhV). We explained the rationale for the selection of reasoning dimensions (Appendix G.1) and tag design (e.g., “Characterization,” “Stability”) (ru54). We provided statistical information of the multi-step SFT data filtering process (Appendix D, Tab.8), confirming the effectiveness of the filtering procedure in improving reasoning quality (Pp95). We expanded the explanations of biochemical jargon and acronyms in the main text, improving its readability and comprehension (6Cmn).

We believe these additions and clarifications have improved the technical rigor, clarity, and completeness of our work, and we will incorporate them into the revised version.

Best Regards,

Authors

---

### Meta-Review · Area_Chair_hT77 · 2026-01-06

**Summary:**

The reviewers agree that the paper presents a novel and well-designed LLM-based framework for drug discovery that explicitly targets interpretability and multi-objective optimization. Key concerns focused on the strength of the interpretability claim, the use of syntactic reasoning rewards, reliance on AutoDock Vina, and the lack of explicit 3D geometric modeling. The authors addressed these issues through careful revisions and additional experiments. They clarified the scope of interpretability, added ablation studies on geometry-related reasoning, introduced alternative evaluation methods (Boltz2 and experimental pIC50), and improved explanations of reasoning design and dataset construction. While the method does not directly model 3D structures, the evidence supports that structured reasoning provides meaningful guidance and consistent performance gains. Overall, the rebuttal resolves the major concerns, thus it is suggested to accept this paper.

**Reviewer Concerns:**

Concerns addressed by rebuttal:

The rebuttal largely addresses several critical concerns, including the interpretability claim, potential evaluation circularity around Vina, and contribution of geometry-related reasoning.

Outstanding concerns:
- The 1D–3D contradiction is acknowledged but not fundamentally resolved
- Interpretability claims still rely heavily on post-hoc reasoning traces rather than verifiable causal explanations (general concern).

**Reviewer Scores:**

The reviewer scores are unlikely to change, as they are already positive on the paper and some minor concerns are not addressed sufficiently.

---

### Decision · Program_Chairs · 2026-01-26

Accept (Poster)